# Cytokines: Can Cancer Get the Message?

**DOI:** 10.3390/cancers14092178

**Published:** 2022-04-27

**Authors:** Rachel M. Morris, Toni O. Mortimer, Kim L. O’Neill

**Affiliations:** Department of Microbiology and Molecular Biology, Brigham Young University, Provo, UT 84602, USA; r.m.morris.9816@gmail.com (R.M.M.); tonmortime@gmail.com (T.O.M.)

**Keywords:** angiogenesis, cytokines, inflammation, tumor microenvironment

## Abstract

**Simple Summary:**

Cytokines are important molecular players in cancer development, progression, and potential targets for treatment. Despite being small and overlooked, research has revealed that cytokines influence cancer biology in multiple ways. Cytokines are often found to contribute to immune function, cell damage, inflammation, angiogenesis, metastasis, and several other cellular processes important to tumor survival. Cytokines have also proven to have powerful effects on complex tumor microenvironment molecular biology and microbiology. Due to their heavy involvement in critical cancer-related processes, cytokines have also become attractive therapeutic targets for cancer treatment. In this review, we describe the relationship between several cytokines and crucial cancer-promoting processes and their therapeutic potential.

**Abstract:**

Cytokines are small molecular messengers that have profound effects on cancer development. Increasing evidence shows that cytokines are heavily involved in regulating both pro- and antitumor activities, such as immune activation and suppression, inflammation, cell damage, angiogenesis, cancer stem-cell-like cell maintenance, invasion, and metastasis. Cytokines are often required to drive these cancer-related processes and, therefore, represent an important research area for understanding cancer development and the potential identification of novel therapeutic targets. Interestingly, some cytokines are reported to be related to both pro- and anti-tumorigenicity, indicating that cytokines may play several complex roles relating to cancer pathogenesis. In this review, we discuss some major cancer-related processes and their relationship with several cytokines.

## 1. Introduction

In 2020, an estimated 19.3 million new cancer cases and nearly 10 million cancer deaths occurred worldwide [1]. Due to cancer’s increasing global impact, understanding factors that drive cancer development has never been more critical. Cancer is characterized by improper regulation of cell differentiation, proliferation, apoptosis avoidance, growth suppressor evasion, increased vasculature, invasion, metastasis, reprogrammed cellular metabolism, and immune evasion [2]. Many of these cancer-promoting processes are highly regulated by cytokines, small protein molecular messengers produced by both normal cells and cancer cells. Cytokines facilitate various interactions between cancer cells, immune cells, and non-immune cells [3,4]. In normal immune activation, cytokines regulate T-cell activation, priming, and CD4+ differentiation and, therefore, play an important role in anticancer immunity [5]. In addition to T cells, cytokines are secreted by other immune cells. For example, tumor-associated macrophages (TAMs), cancer-associated fibroblasts (CAFs), and myeloid-derived suppressor cells (MDSCs) secrete chemokines C–C motif chemokine ligand 2 (CCL2), CCL4, and CCL5 and inflammatory cytokines tumor necrosis factor α (TNF-α), transforming growth factor β (TGF-β), interleukin-1β (IL-1β), IL-6, and IL-23 to activate T helper 17 (Th17) cell expansion [6]. Generally, pro-inflammatory cytokines mediate key immune interactions to promote antitumor activity. In contrast, cytokines secreted by cells in the tumor microenvironment (TME) and some normal cells promote various cancer processes, such as angiogenesis, epithelial to mesenchymal transition (EMT), invasion, tumor progression, and maintain cancer stem-cell-like (CSCs) cells [7]. There are several cytokines involved in these processes. Some cytokines even have complex dual roles and may be involved in both immune activation and cancer development, while others are undergoing investigation to become novel therapeutic targets to treat cancer by disrupting cancer processes. Overall, cytokines are heavily involved in multiple aspects of cancer development and may drive carcinogenesis or promote antitumorigenic effects. Here, we discuss the complex relationship between cytokines and various cancer-related cellular processes, such as immune activation and suppression, inflammation, cell damage, angiogenesis, CSC maintenance, invasion, and metastasis.

## 2. Cytokines Regulate Key Immune Players

Macrophages are members of the innate immune response and are responsible for phagocytosis of foreign materials, engulfment of dead, injured, or infected cells, extracellular matrix formation, angiogenesis, and antigen presentation [8]. Macrophages can be divided into two functional classes: Macrophages 1 (M1) and Macrophages 2 (M2). M1 macrophages can switch to become M2-type macrophages, or vice versa, depending on environmental conditions, such as inflammation, infection, hypoxia, injury, or cytokine secretion [8]. However, the distinction between M1 and M2 macrophages is a simplified view of macrophage polarization and is better represented as a continuum of macrophage functional states [9]. Macrophages alter their behavior to address environmental stressors. TNF-α, IFN-γ, IL-12, IL-23, Toll-like receptor (TLR) ligands, and LPS promote the M1 phenotype, while IL-14 and IL-13 induce the M2 phenotype [8,10,11]. Simply, M1-type macrophages are generally pro-inflammatory, activate Th1 cell responses, and inhibit cell proliferation via tissue damage caused by the secretion of pro-inflammatory cytokines. In contrast, M2 macrophages favor immunosuppressive behavior, poor antigen presentation, wound healing, angiogenesis, cell proliferation, promote Th2 cell activity, and suppress Th1 activity (Figure 1) [8]. M2 macrophages secrete C–X–C motif chemokine ligand 13 (CXCL13), which promotes cell proliferation, invasion, migration, and EMT in renal cell carcinoma (ccRCC) cells by interacting with CXCR5 receptors [12]. TAMs generally have M2 functions and promote immunosuppression by releasing CCL18 to attract naïve T cells lacking cytotoxicity to the TME. CCL18 from TAMs is associated with increased breast cancer metastasis and poor prognosis [13]. CCL17 and CCL22 are additional TAM chemokines that interact with CCR4 receptors displayed on regulatory T cells (Treg) and Th2 cells, which are T-cell subsets devoid of antitumor activity [9]. TAMs represent over 50% of tumor-infiltrating immune cells and, therefore, may act as prominent players in regulating cancer development [14].

Th17 cells represent one T-cell subset reported to have both anti- and pro-tumor effects. Depending on tumor type and conditions, Th17 cells may drive chronic inflammation, which is associated with several tumor types, or enhance the recruitment of CD8+ T cells, neutrophils, and natural killer (NK) cells to the tumor and activate tumor killing [15]. Th17 cell clonal expansion and differentiation require a wide variety of cytokines, such as TGF-β, IL-1β, IL-6, IL-21, and IL-23 [16,17,18]. Th17 cells were first distinguished as different from other CD4+ T cells when retinoic acid receptor-related orphan receptor gamma t (RORγt) was identified as a transcriptional factor required for Th17 cell differentiation [19]. Type 3 cytokines IL-17A, IL-17F, IL-22, and IL-26 are secreted by Th17 cells [20]. The IL-17 superfamily comprises cytokines IL-17A-F and receptors IL-17RA-RE. Little is known about the functions of IL-17B, -C, and -D; however, IL-17A and IL-17F are highly involved in inflammation and autoimmunity. IL-17E, also known as IL-25, promotes Th2 cell-specific cytokines while inhibiting Th17 cell development [21]. Th17 cells secrete IL-17 to guide neutrophils and macrophages to the tumor site and promote anticancer activity [6]. IL-17 also stimulates production of inflammatory cytokines IL-6, colony-stimulating factors (G-CSF and GM-CSF), IL-1β, and chemokines CXCL2 and CXCL8 to promote granulocyte recruitment in response to inflammation [22].

Foxp3-expressing Tregs play key roles in preventing autoimmunity via self-tolerance; however, evidence suggests that they facilitate cancer progression by suppressing immune cells and regulating immune surveillance [23]. Increased Treg activity is associated with cancer progression in multiple tumor types in humans [24]. A recent study observed Treg increases with lymph node invasion in breast cancer [25]. Another study revealed that lymph nodes containing metastatic breast cancer have increased Treg frequency, effector T-cell exhaustion, and suppressed TCR signaling, compared with non-metastatic lymph nodes [26]. Cytokines and environmental cues direct Treg responses to inhibit inflammation [27]. Tregs may secrete TGF-β, IL-10, and IL-35 to inhibit immune activation [28]. One study showed that CD8+ T-cell suppression is linked to TGF-β signaling [29]. Interestingly, TGF-β drives Treg cell differentiation; however, when IL-6 or IL-21 is also present, Treg cell differentiation is inhibited, while Th17 cell differentiation is promoted, demonstrating cytokine-dependent Treg/Th17 plasticity [30]. In breast cancer, Tregs are reported to upregulate chemokine receptor CCR8 [24]. Elevated CCR8 discriminates highly suppressive tumor Tregs from systemic lymphoid tissue Tregs. Interestingly, CCR8 is not required for Treg-driven immunosuppression and Treg cell accumulation in mouse models [31]. Kidani et al. showed that administering anti-CCR8 monoclonal antibodies (mAb) to tumor-bearing mice results in CD8+ effector T-cell expansion, less T-cell exhaustion, and tumor immunity when mice were rechallenged with the same tumor cell line. Results also indicated that anti-CCR8 mAbs promote minimal autoimmunity compared to systemic Treg depletion [32]. Therefore, CCR8 is a potential therapeutic target to selectively remove tumor Tregs and avoid systemic Treg removal and autoimmunity

## 3. Cytokines in Inflammation and Cellular Damage

Inflammation, rather than a singular event, is a process involving cells of innate and adaptive immunity being activated, recruited, and put into action. Inflammation is essential for host defense against pathogens, as well as tissue repair and regeneration [33,34]. During the early onset of an inflammatory response, neutrophils migrate to the inflamed site, followed by leukocytes, lymphocytes, and other activated inflammatory cells, which are attracted by a signaling network of cytokines, chemokines, and growth factors [35].

Cytokines are largely involved in inflammation and have specific effects on the communications and interactions between cells. Through autocrine, paracrine, or endocrine action, cytokines may act on the cells that secrete them, neighboring cells, or distant cells [36]. By elaborate biological mechanisms, the production and action of cytokines are closely regulated. Under normal healthy conditions, cytokines are produced and consumed at the site of tissue damage or invasion [37]. Various pro-inflammatory cytokines, such as IL-1β, IL-6, and TNF-α are shown to influence the process of pathological pain as well as the upregulation of inflammatory reactions [36]. While modest systemic levels of these cytokines may bring benefits in promoting mild fever, mobilization of hemopoietic progenitors, and protein production, marked elevation of IL-1β, IL-6, and TNF-α in severe disease can contribute to tissue damage and organ failure [37,38,39]. Inflammation appears to be an important factor in the ability of tumor cells to metastasize. The presence of inflammatory cells and mediators in the TME is a hallmark of cancer-related inflammation [40]. TAMs are a major source of cytokines in the TME. The nutrient-depleted and hypoxic TME provokes M2-type macrophage activation, which, in turn, aids tumor progression and suppresses antitumor activity. Hypoxia also induces increased IL-10 production in macrophages, supporting alternate activation [41]. Through the secretion of pro-inflammatory cytokines such as IL-6, IL-1β, and TNF-α, macrophages can contribute to tumor-promoting inflammation [42]. Due to this, it is crucial that cytokines are regulated to ensure that they stimulate their proper target cells and inhibit excessive inflammation [37]. Self-limiting characteristics of an acute inflammatory response allow for inflammation to resolve under normal circumstances. Various anti-inflammatory mediators are involved in the negative regulation of inflammation, including cytokines IL-10 and TGF-β. A balance between inflammatory and anti-inflammatory cytokines is necessary for maintaining normal physiology.

IL-10 is a potent anti-inflammatory cytokine needed for the suppression of pro-inflammatory IL-17 expressing Th17 cells, as well as the regulation of anti-inflammatory Tregs. Evidence that IL-10 controls tumor-promoting inflammation is reported in several studies. One study showed that IL-10 knockout mice develop colon cancer. Other studies indicated that IL-10 deficiency in humans correlates with lymphoma development [43,44,45]. Early clinical trials focused on the anti-inflammatory functions of IL-10. Patients with various inflammatory diseases received injections of IL-10. Results showed significantly reduced levels of disease-associated pro-inflammatory cytokines, such as TNF-α, IL-1β, IL-12, and IL-17; however, this reduction was dependent on the continuous elevation of IL-10 [43,46,47].

Nitrative and oxidative DNA damage from reactive oxygen species (ROS) and reactive nitrogen species (RNS), such as nitric oxide (NO), is also related to various inflammatory conditions and a precursor to carcinogenesis. Infection and inflammation trigger NO generation by inducible nitric oxide synthase (iNOS) [48]. iNOS induction is closely associated with cytokines TNF-α, IL-1β, and IL-6, which are mediators of innate immunity. While iNOS induction by cytokines represents a rapid mechanism of host defense against some pathogens, it is less antigen-specific than the adaptive immune response [49]. Hypoxia-inducible factor-1α (HIF-1α) mediates transcription of iNOS. To adapt to an hypoxic environment, tumor cells increase the synthesis of HIF-1α. One study explored the relationship between iNOS and HIF-1α and their correlation with nitrative and oxidative DNA damage, such as 8-nitroguanine and 8-oxo-7,8-dihydro-2′-deoxyguanosine (8-oxodG). The results of the study showed that iNOS expression is mediated by the expression of HIF-1α and results in DNA damage [50]. Increased expression of HIF-1α is shown to result from sustained exposure to inflammatory cytokines, such as IL-1β, IL-6, and TNF-α [51]. Further, 8-oxodG damage leads to G→T transversions [52], which are frequently found in several tumor-relevant genes. These damage events activate DNA damage response (DDR) signaling, including lesion repair and cell death. The activation of DDR-driven pro-inflammatory signals, such as NF-κB or various interleukins, can lead to chronic inflammation [53].

Uncontrolled forms of inflammation, such as chronic inflammation and cytokine storm, may influence the development of disease and disorders. When tissue inflammation becomes chronic, a greater risk arises for the development of malignancies. This correlation between inflammation and cancer was suggested in 1863 by Rudolph Virchow, who discovered leukocytes in human breast carcinomas [34,40]. Since then, epidemiological data have accumulated supporting Virchow’s original hypothesis of inflammation-mediated oncogenesis. Such evidence suggests that chronic inflammation and infection are associated with approximately 25% of all human cancers worldwide [34].

## 4. Cytokines Drive Angiogenesis

Angiogenesis, also termed neovascularization, is blood vessel development from pre-existing vasculature. It is regulated by a careful balance of pro- and anti-angiogenic factors. Normal wound healing, tissue regeneration, and physiological growth are associated with angiogenesis; however, unbalanced angiogenesis is a prerequisite for solid tumor development and, therefore, an important research area [54,55]. Tumor vasculature is characterized by an abnormal basement membrane, discontinuous and irregularly shaped endothelium, and tortuous hyperpermeable blood vessels that leak nutrients to support cancer growth, development, and resistance [56]. Tumor blood vessels are disorganized and difficult to identify as arterioles, venules, or capillaries [57]. A diverse group of cells, including endothelial cells, cancer cells, cancer-associated adipocytes, fibroblasts, neutrophils, and macrophages influence tumor vasculature morphology and functionality (Figure 2) [58,59,60,61]. For example, TAMs promote angiogenesis by secreting various pro-angiogenic factors, such as vascular endothelial growth factor (VEGF), metalloproteinases (MMPs), TGF-β, platelet-derived growth factor (PDGF), and adrenomedullin (ADM) [11]. Overall, angiogenesis is a complicated process that involves a diverse group of cells that secrete various angiogenic factors and cytokines. Here, we discuss some major cytokines that are specifically involved in angiogenesis.

### 4.1. Angiogenesis and the VEGF Family

The VEGF family includes VEGF-A, -B, -C, -D, and placental growth factor (PIGF). VEGF-A is known to bind VEGF receptors 1 and 2, while VEGF-B and PIGF bind VEGF receptor 1 only [62]. Under oxygen-deprived conditions, hypoxia-inducible factors (HIF-1α and HIF-2α) induce TAMs to increase VEGF-A secretion and, thus, promote angiogenesis [63]. CAFs are major stromal resident cells that secrete several aggressive angiogenic factors, including FGF2, FGF7, IL-6, and VEGF-A [64]. Several studies indicate that VEGF regulates angiogenesis in several cancer types. For example, VEGF serum levels are correlated with vascular invasion, metastases, tumor stage, and tumor grade of bladder cancer [65]. One study indicated that TAMs overexpressing VEGF-C promoted lymphovascularization in Merkel cell carcinoma [66,67]. As mentioned previously, MMPs also play a major role in driving angiogenesis and may be related to VEGF levels. Early chicken chorioallantoic membrane models and mouse models demonstrated that MMP-2 downregulation and deficiencies are associated with tumor angiogenesis [68,69]. Interestingly, MMP-9- and MT1-MMP-deficient mice present decreased angiogenesis, compared with wild type [70,71,72]. One study proposed that MMP-2 may modify VEGF expression after observing reduced VEGF expression in A549 lung cancer xenograft tissue samples from mice treated with MMP-2 siRNA [73]. Several studies demonstrate that MMPs enhance VEGF production; however, the exact mechanism is unknown [74]. In addition to angiogenesis in solid tumors, VEGF may be associated with hematological malignancies [75]. Recently, Filipiak et al. used an enzyme-linked immunosorbent assay (ELISA) to measure VEGF-A levels in 42 Hodgkin’s lymphoma patients and found that VEGF-A concentration was significantly elevated compared to healthy patient levels [76]. Another study showed that treating Raji cells with oxacetaxine and curcumin, which have antiproliferative effects on lymphoma cells, results in suppressed VEGF-A levels in exosomes derived from Raji cells, and decreases phosphorylated VEGF receptor 2 (p-VEGFR2) levels [77]. VEGF may also be used to predict the prognosis of diffuse large B cell lymphoma (DLBCL) patients. Sang et al. performed a retrospective study and discovered that upregulated VEGF was related to poor therapeutic response and survival of DLBCL patients [78]. In summary, extensive research has identified VEGF as an angiogenesis biomarker and potential therapeutic target.

A recent preclinical study showed that propofol, a common intravenous anesthetic, could inhibit VEGF/VEGFR2- and mTOR/eIF4E-mediated signaling pathways to induce anti-angiogenic activity [79]. However, the relationship between propofol and angiogenesis will require further investigation because both anti-angiogenic and pro-angiogenic effects are reported [80,81]. Anti-VEGF antibodies may target VEGF and inhibit angiogenesis. For example, humanized mAb bevacizumab targets VEGF-A isoforms and was approved in the US in 2004 to be used in combination with chemotherapy for colorectal cancer treatment. Today, it is also approved to treat ovarian, cervical, glioblastoma, and renal cancers [65]. Bevacizumab is also being tested as a potential treatment option for other cancers (Table 1). In one case study, a 21-year-old female with central nervous system (CNS) acute myeloid leukemia (AML) relapse was treated with bevacizumab and intrathecal (IT) chemotherapy. As a result, she remained in complete remission for nearly 1 year. Results suggest that bevacizumab may prove to be a good combination treatment for AML patients [82].

PIGF is a potential prognostic cancer biomarker, like other VEGF family members, and is involved in endothelial stimulation, bone marrow-derived cell activation, and angiogenesis [83]. In one study, the serum levels of PIGF from 49 clear cell ccRCC patients were tested before surgery and 3 months post-surgery. Before surgery, patients with primary metastatic ccRCC had significantly elevated PIGF levels, compared with localized, without-relapse ccRCC patients [84]. Phage-display technology identified PIGF-specific nanobodies in the nM range that demonstrate anti-angiogenic activity [85,86]. Humanized anti-PIGF antibody TB-403 was tested in patients with advanced solid tumors in phase I clinical trial and pediatric patients with relapsed or refractory medulloblastoma (NCT02748135) [87]. Results are expected to be released in 2022 [88].

### 4.2. Angiogenesis and Other Major Cytokines

Hepatocyte growth factor (HGF) is a scatter factor that promotes EMT, by activating metalloproteinases, and potently binds to heparin-binding angiogenic factor [89]. HGF is a stromal-cell-derived cytokine and the natural endogenous ligand for mesenchymal–epithelial transition (MET) receptor tyrosine kinase (RTK), which is encoded by the MET proto-oncogene on human chromosome 7 [90]. HGF/MET signaling activation drives angiogenesis, cell proliferation, and tumor aggressiveness [91]. Elevated levels are associated with poor colorectal and lung cancer survival [92,93]. Recently, Katayama et al. revealed that high HGF serum levels in muscle-invasive bladder cancer (MIBC) patients are associated with worse cancer-specific survival, recurrence-free survival, and overall survival [94]. Another study found that patients with gastric cancer have elevated plasma HGF levels, compared with patients with normal gastric mucosa or gastric ulcers [95]. Developing novel treatments targeting both HGF-dependent and HGF-independent MET activation will be important for future treatments targeting the oncogenic-driving MET pathway [96]. 

Interleukins are involved in multiple cancer processes, including angiogenesis. The IL-1 family comprises IL-1α, IL-1β, and IL-1 receptor antagonist (IL-1RA). IL-1α and IL-1β are cancer-promoting and drive angiogenesis, tumor progression, and tumor aggressiveness [97]. CAFs and adipocytes are known to promote these processes through IL-1β secretion [98]. Early studies of B16 melanoma mouse models showed that IL-1β is required for angiogenesis via lymphotoxin and VEGF-A induction, while IL-1α promoted a similar but weaker phenotype [99,100]. Endothelial cells act as both direct and indirect targets of IL-1 signaling and produce VEGF upon activation (Figure 3). IL-1 and VEGF may synergize to enhance angiogenesis [98]. IL-1β/IL-1R1 signaling induces additional IL-1β transcription in macrophages, which promotes fibroblasts and endothelial cells to also contribute to angiogenesis [101]. Recently, Machelke et al. found that epidermal growth factor receptor (EGFR) inhibition in A549 lung cancer cells leads to reduced IL-1β-induced tissue factor (TF) expression, which normally promotes tumor progression and angiogenesis [102]. The Canakinumab Anti-inflammatory Thrombosis Outcomes Study (CANTOS) trial evaluated anti-IL-1β to treat atherosclerosis, and results indicated that treatment significantly reduced lung cancer incidence [97,103].

Other interleukins, such as IL-6 and IL-8, may also mediate angiogenesis. Early studies showed that serum IL-6 and VEGF levels are related. IL-6 may induce VEGF expression to increase vasculature [104]. IL-6 is known to transmit signals through several signaling pathways, including JAK/STAT, RAS/MAPK, PI3K/Akt, and NF-κB, to drive tumor progression [105]. IL-6 binds to the IL-6 receptor (IL-6R), which results in the release of its associated Janus kinase (JAK) to phosphorylate transcription 3 (STAT3) and initiates downstream signals to promote angiogenesis, proliferation, and prevent apoptosis [106]. Li et al. demonstrated that IL-6 increases angiogenesis via the STAT5/P-STAT5 signaling pathway and that 6-phosphofrutcto-2-kinase/fructose-2, 6-bisphosphatase 4 (PFKFB4) expression elicits IL-6 upregulation via NF-κB signaling to increase breast cancer angiogenesis [107]. Several potential treatment strategies to target IL-6 are currently undergoing investigation. Examples include (1) small molecule Madindoline A to inhibit dimerization of IL-6/IL-6R/gp130 complexes, (2) siltuximab and CNTO-136 to inhibit IL-6 activity, and (3) mAb tocilizumab to block IL-6R [106]. IL-8 is a chemokine that binds C–X–C motif chemokine receptor 1 (CXCR1) and CXCR2, which are G-protein coupled receptors displayed on granulocytes, monocytes, and endothelial cells to increase angiogenesis, recruit immunosuppressive cells to the tumor site, and worsen prognosis [108]. IL-8 was demonstrated to be the primary cytokine involved in increasing endothelial cell permeability and cell junction disruption in glioblastoma [109]. In human colorectal cancer mouse models, IL-8 induces significant increases in CD31+ peritumoral vasculature, while CXCR2 knockout results in significantly reduced tumor growth, potentially due to lack of IL-8 signaling [110]. Interestingly, gastric cancer cells treated with nicotine, an alkaloid found in tobacco, show enhanced angiogenesis and proliferation in the TME by stimulating IL-8 expression via ROS/NF-κB and ROS/MAPK (Erk1/2, p38)/AP-1 pathways [111]. Like other interleukins, IL-8 may also be targeted for cancer treatment. Recently, escin, a pentacyclic triterpenoid derived from horse chestnut, demonstrated antitumor activity against pancreatic cancer cells by influencing IL-8 expression. Results indicated that escin-treated pancreatic cancer cells had significantly reduced NF-κB activity and IL-8 and VEGF secretion, resulting in inhibited angiogenesis [112].

The angiopoietin family consists of four members that bind to Tie-2 receptors on endothelial cells [89]. Generally, Ang-1 is described as a strong Tie-2 agonist, while Ang-2 is a Tie-2 antagonist that competes with Ang-1. However, recent studies characterize Ang-2 as capable of acting as a Tie-2 agonist or antagonist [113]. In colorectal cancer, Ang-2 expression is negatively associated with patient overall survival [114]. One study demonstrated that glucocorticoid-treated colon cancer-derived myofibroblasts reduce Ang-2 levels and inhibit endothelial cell angiogenesis and cell migration [115]. Ang-4 and Ang-3, a mouse orthologue, are both Tie-2 agonists, but their effect on angiogenesis is not well characterized [116]. A recent study showed that Ang-3 was elevated in cervical cancer cells, compared with normal cervical cells, and Ang-3 silencing inhibited human umbilical vein endothelial cell angiogenesis and integrin alpha v beta 3 (αvβ3). Results also showed that upregulated αvβ3 expression increases VEGF and VEGFR2 secretion and blood vessel formation, suggesting Ang-3 as a potential novel therapeutic target for treating cervical cancer [117]. In another study, Ang-4 was discovered to be overexpressed in ovarian cancer cells. Immunoprecipitation results suggested that Ang-4 suppression leads to VEGFR2/VE-cadherin/Src complex dissociation and phosphorylation of VEGFR2 in A2780 and CAOV3 ovarian cancer cell lines. Researchers concluded that Ang-4 silencing significantly inhibits tumor angiogenesis and progression [118]. 

## 5. Cancer Stem Cells, Cellular Plasticity, and Cytokines

CSCs represent a small population of cells within a tumor that have unlimited proliferation, differentiation, and self-renewal abilities. Generally, CSCs are believed to be highly responsible for treatment failure because they are progenitor cells that may survive conventional treatment and replenish the tumor. They divide into heterogeneous cancer cell types that promote treatment resistance and cancer recurrence [119]. Cellular plasticity allows cancer cells to dynamically shift between a differentiated state and an undifferentiated or CSC state. Examples of cancer cell plasticity in action are EMT, invasion, and metastasis (Figure 4) [120]. Cytokines derived from stromal cells or immune cells are known to activate stemness and promote immune evasion and, consequentially, drive non-CSCs to become more CSC-like [121]. Macrophages that infiltrate the TME secrete cytokines, such as IL-6, IL-10, TNF-α, and TGF-β, which may enhance cancer cell stemness and EMT [122]. Wan et al. observed that TAMs cocultured with HCC stem cells secrete IL-6 to activate STAT3 signaling, promote sphere formation, and increase CD44+ HCC cells [123]. TNF-α is a pro-inflammatory cytokine released by activated immune cells to elicit antitumor activity; however, some recent studies have discovered that TNF-α may play a dual role and also promote tumor progression [122]. TNF-α is reported to increase EMT and CSC transition in various tumor cell types and increase cancer transformation, proliferation, and angiogenesis [124,125,126]. In colon cancer, TNF-α activates PI3K/Akt and p38 MAPK parallel signaling pathways to induce CXCL10 transcription, a pro-inflammatory cytokine that binds to CXCR3, which further leads to EMT increase [127]. In ccRCC, TNF-α promotes stemness and EMT through MMP-9 activation and the PI3K/AKT/GSK-3β signaling pathway. Inhibiting PI3K/AKT reactivates GSK-3β and prevents ccRCC cells from undergoing EMT by suppressing TNF-α [128]. TNF-α antagonists may be used to neutralize TNF-α and reprogram the TME to become more antitumor-like [129,130].

Other pro-inflammatory cytokines may also regulate CSCs. One study showed that colon cancer CSCs increase when myofibroblasts secrete IL-6 and IL-8 and activate the Notch pathway through STAT3 [131]. Notch signaling is highly upregulated in colon cancer CSCs, and regulates the self-renewal abilities of a cell, prevents apoptosis, and suppresses cell lineage differentiation genes [132]. In liver cancer, Liu et al. showed that IL-6, TGF-β, and monocyte chemoattractant protein 1 (MCP-1) levels are upregulated. Further investigation revealed that inhibiting these pro-inflammatory cytokines significantly suppresses liver cancer growth and promotes apoptosis. The study also indicated that liver cancer stem cells markers, CD90 and CD133, were positively correlated with pro-inflammatory factors [133]. Recently, magnolol, a bioactive polyphenolic component that exhibits anticancer properties, was administered to oral squamous cell carcinoma CSCs. Results indicated that magnolol targets CSCs and suppresses stemness, self-renewal, and cell viability by downregulating IL-6/STAT3 signaling [134]. Overall, CSCs are an important tumor component and are highly regulated by cytokines.

## 6. Cytokines in Cancer Invasion and Metastasis

Metastasis involves complex processes that endow malignant cells with the ability to survive and grow within the primary TME, migrate to and invade other tissues, and colonize target organs through establishing disseminated cells at and around the target site [135,136]. Chronic deregulation of cytokine activity and expression is associated with the risk of disease. Literature suggests that several cytokines are involved in signaling to cancer cells and supporting the growth and survival of tumor cells, as well as the enhancement of other metastatic properties [137].

IL-1 expression is elevated in breast cancer and is suggested to be involved in metastasis. Specifically, it is thought that IL-1β is secreted into the tumor microenvironment, activating inflammation and promoting invasion [138]. A study by Soria et al. suggests that coordinated expression of TNF-α and IL-1β promotes invasion and disease relapse. TNF-α mainly induces EMT, a process essential for tissue repair but also associated with invasion and metastasis. The upregulation of both TNF-α and IL-1β was observed in breast cancer patients with reoccurring disease [139]. IL-6 is another cytokine involved in metastatic dissemination. While IL-6 plays an important role in multiple physiological processes including cell proliferation, acute inflammation, and metabolism, active IL-6/JAK/STAT3 signaling drives cancer cell proliferation and invasiveness and suppresses apoptosis. IL-6 is often detected at increased levels in breast cancer patients [140]. Various other cytokines are believed to be involved in the metastasis and progression of several cancer types through similar pro-inflammatory activity, including IL-4, IL-8, IL-10, and TGF-β [137,138,141]. 

A critical step in the metastatic cascade is the development of enhanced cell motility, allowing tumor cells to invade adjacent tissue [136]. Initially, a breach in the basement membrane barrier and dissociation of tumor cells from the primary tumor occurs. Adjacent tissue is invaded, and intravasation and extravasation from vasculature carry tumor cells to a secondary anatomical site where the colonization of a target organ occurs [135,136]. Normally, displaced cells are efficiently removed by anoikis, which causes apoptotic death due to loss of adhesion [142]. For cancer to successfully metastasize, it is important that the anoikis be inhibited [143]. Once EMT is initiated, several epithelial proteins are lost, including E-cadherin, β-catenin, and γ-catenin, and an increase in mesenchymal proteins, such as N-cadherin, vimentin, and fibronectin, occurs. Upregulation of transcriptional repressors of E-cadherin is one of the hallmarks of E-cadherin loss and EMT. TGF-β induces EMT, upregulates these transcriptional factors, and promotes metastasis by inducing anoikis resistance due to a loss of E-cadherin and increasing the N-cadherin expression. Anoikis resistance is stimulated by several cytokines through the activation of survival pathways, including the IL-6/STAT3 pathway [144,145]. There are several other mechanisms for anoikis resistance that contribute to metastasis [146]. 

Metastasis and inflammation are also regulated by death ligands such as tumor necrosis factor-related apoptosis-inducing ligand (TRAIL; TNFSF10) and corresponding death receptors. The TRAIL/death receptor signaling pathway plays both negative and positive roles in regulating cancer invasion and metastasis. In humans, there are two TRAIL death receptors: death receptor 5 (DR5; also called TRAIL-R2 or Killer/DR5) and death receptor 4 (DR4; also called TRAIL-R1). Under normal conditions, TRAIL ligation with its death receptors (DR4 and DR5) on the surface of cancer cells induces the formation of the death-inducing signaling complex (DISC) involving Fas-associated death domain (FADD) recruitment of pro-caspase-8 via its death effector domain, resulting in caspase-8 or -10 activation, followed by cleavage and activation of caspase-3, -6, and -7, and eventual execution of apoptosis or anoikis. Inhibition of TRAIL/death receptors causes available FADD and caspase-8 to recruit and stabilize TNF-receptor-associated factor (TRAF) 2, resulting in enhanced TRAF2 polyubiquitination and activation. This leads to activation of ERK/JNK/AP-1 signaling and NF-κB activation, subsequently activating MMPs and enhancing the release of inflammatory cytokines that promote invasion and metastasis of cancer cells [147]. Through many complex mechanisms and signaling pathways, chronic dysregulation of cytokine activity contributes to cancer invasion and metastasis.

## 7. Conclusions

Due to cancer’s global impact, understanding how cancer develops and progresses is critical. Cytokines are heavily involved in regulating cancer developmental processes. Immune activation and suppression, inflammation, cellular damage, angiogenesis, CSCs, invasion, and metastasis are all related to cancer and controlled by cytokines. Cytokines have profound effects on cells and may lead to pro- or antitumor activity, depending on environmental conditions and the presence of other cytokines. Overall, studies have shown that cytokines are small messengers that are often overlooked as powerful molecular players in the TME. The impact and therapeutic potential of cytokines are beginning to be uncovered through already existing and developing cytokine-targeted therapies. In this review, we highlighted only a few cytokines and cancer-related processes. Further research focusing on the intricate relationship between cancer and cytokines will be required to increase our understanding of how cancer behaves. Cytokine research will continue to play a key role in revealing the molecular mechanisms behind cancer development and identifying novel therapeutic targets.

## Figures and Tables

**Figure 1 cancers-14-02178-f001:**
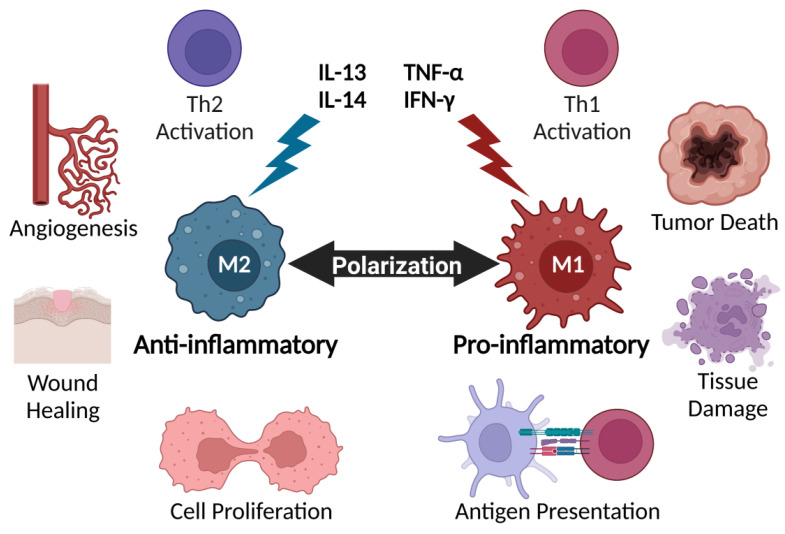
Macrophage polarization is highly regulated by cytokines. Generally, inflammatory cytokines promote M1-type macrophage functions that lead to enhanced Th1 activation, tumor killing, tissue damage, and antigen presentation. In contrast, anti-inflammatory cytokines induce macrophages to become more M2-like, leading to increased tumor-promoting and immunosuppressive responses, such as Th2 activation, angiogenesis, wound healing, and cell proliferation.

**Figure 2 cancers-14-02178-f002:**
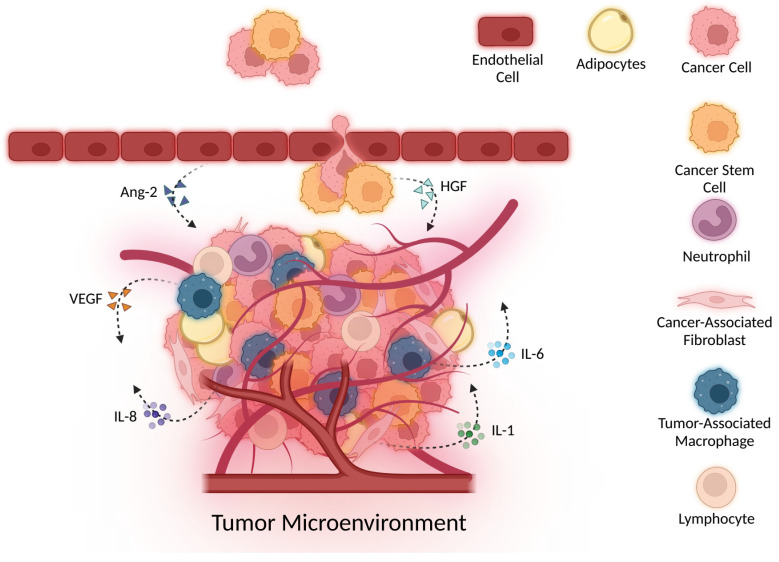
The TME is composed of a diverse group of cells, including cancer cells, CSCs, adipocytes, neutrophils, TAMs, CAFs, and lymphocytes. Various cytokines secreted by these cells enhance angiogenesis and overall tumor progression.

**Figure 3 cancers-14-02178-f003:**
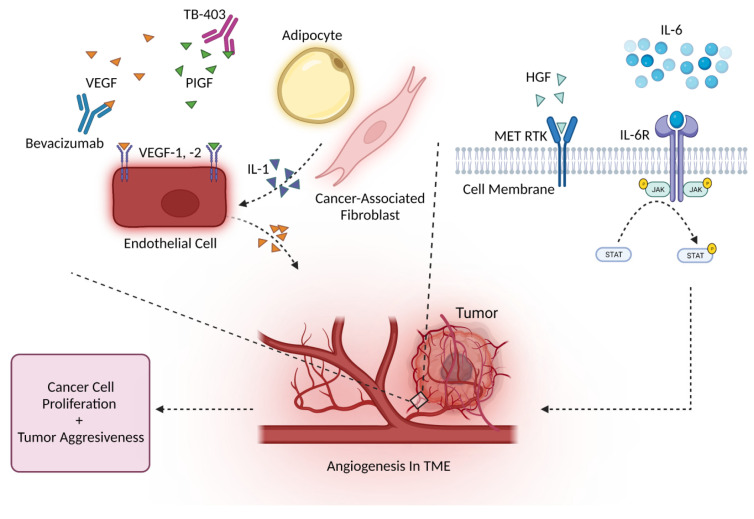
Several cytokines, cell types, and signaling pathways drive angiogenesis. Increased vasculature supports tumor growth and aggressiveness. Bevacizumab may be used to block VEGF, while TB-403 may potentially be used to inhibit PIGF to further prevent angiogenesis in TMEs.

**Figure 4 cancers-14-02178-f004:**
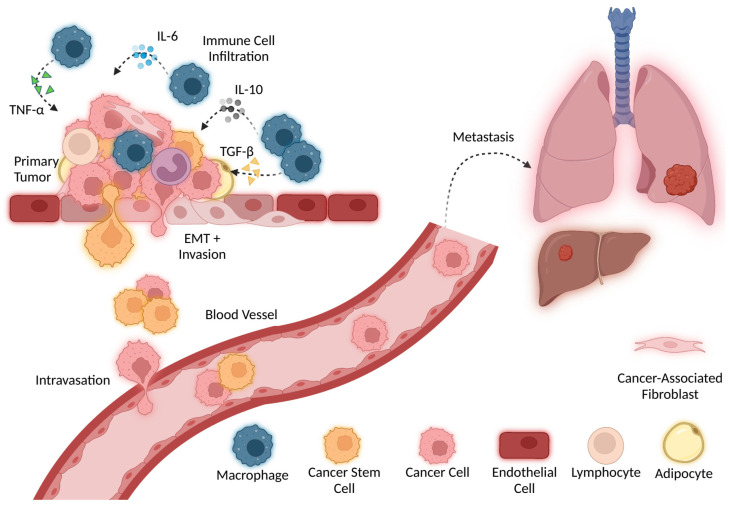
CSCs and cytokines secreted by immune cells infiltrating the tumor promote EMT and invasive activity. Eventually, cancer cells may intravasate into local blood vessels to metastasize and colonize other tissues.

**Table 1 cancers-14-02178-t001:** Representative clinical trials: cytokine targeted immunotherapy strategies in monotherapy or combination (as of April 2022) may prove promising for the treatment of various cancers.

Cytokine	Intervention/Treatment	Phase	Clinical Trial
TGF-β	Galunisertib + nivolumab Galunisertib + durvalumab Bintrafusp alfa + radiation therapy Fresolimumab + radiotherapy	Phase 1, 2 Phase 1 Phase 1 Phase 1, 2	NCT02423343 NCT02734160 NCT03524170 NCT02581787
TNF-α	Nivolumab + ipilimumab + certolizumab or infliximab L19 TNF-α + doxorubicin	Phase 1 Phase 1	NCT03293784 NCT02076620
IL-2	NKTR-214 + pembrolizumab NKTR-214 + nivolumab NKTR-214 + nivolumab + ipilimumab aldesleukin Aldesleukin + bevacizumab Atezolizumab + cergutuzumab amunaleukin RO6874281 + trastuzumab + cetuximab	Phase 1, 2 Phase 1 Phase 2 Phase 2 Phase 4 Phase 2 Phase 1 Phase 1	NCT03138889 NCT02983045 NCT03282344 NCT00006864 NCT00853021 NCT02350673 NCT02627274
IL-10	Pegilodecakin + FOLFOX	Phase 3	NCT02923921
IL-15	N-803 rhIL-15 + NK cell infusion N-803 + aNK (NK-92)	Phase 2 Phase 1 Phase 2	NCT02989844 NCT01875601 NCT02465957
IL-12	Electroporated plasmid + IL-12p DNA Pembrolizumab + pIL-12	n/a Phase 2 Phase 2	NCT00323206 NCT02345330 NCT02493361
IL-8	BMS-986253 + nivolumab or nivolumab + ipilimumab	Phase 1, 2	NCT03400332
VEGF	Bevacizumab + atezolizumab or sunitinib	Phase 2	NCT01984242
CSF-1	APX005M + cabiralizumab + nivolumab pexidartinib + durvalumab LY3022855 + durvalumab or tremelimumab	Phase 1 Phase 1 Phase 1	NCT03502330 NCT02777710 NCT02718911
GM-CSF	Docetaxel and GM-CSF	Phase 2	NCT00488982
PIGF	TB-403	Phase 1	NCT02748135

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
