# Peer review of "Cytokines: Can Cancer Get the Message?"

_cancers, 2022, doi:10.3390/cancers14092178_

Round 1

Reviewer 1 Report

Though cytokines play an important role in chronic inflammation sometimes leading neoplastic diseases, their complex role in these processes is still unfolding. Further the cytokines control almost all aspects of immunosurveillance. The tumor microenvironment consists of numerous types of immune cells and enzymes with intricate signaling network which is regulated by cytokines. This review gives a comprehensive picture of the role of cytokines in the interphase between the development of cancer and immune system. Further, the review provides information on the relationship between inflammation and neoplastic development focusing on the role of cytokines. I would like to see two more illustrations (figures) for cancer stem cells, tumor microenvironment and their contribution towards neoplastic development. 

Author Response

“I would like to see two more illustrations (figures) for cancer stem cells, tumor microenvironment and their contribution towards neoplastic development.”

Thank you for this wonderful suggestion. We have produced two new figures as requested. The first new figure illustrates the complex nature of the tumor microenvironment, while the second new figure shows how cancer stem cells and cytokines contribute to epithelial–mesenchymal transition, invasion, and metastasis. The new figure legends have been added to the bottom of the paper and are included below in red. Incorporating this suggestion has greatly enhanced the paper.

Figure 2. The TME is composed of a diverse group of cells, including cancer cells, CSCs, adipocytes, neutrophils, TAMs, CAFs, and lymphocytes. Various cytokines secreted by these cells enhance angiogenesis and overall tumor progression.

Figure 4. CSCs and cytokines secreted by immune cells infiltrating the tumor promote EMT and invasive activity. Eventually, cancer cells may intravasate into local blood vessels to metastasize and colonize other tissues.

Reviewer 2 Report

Manuscript entitled “Cytokines, Can Cancer Get The Message?„ is comprehensive review, well written article with topic related to progression and growth of tumor due to changes in microenvironment and inflammatory cell profile. Cytokines are involved in regulation of pro-tumoral activities and in this article most important are mentioned and the molecular pathways are correlated.

This article would be of even better quality with the addition of one paragraph that would refer to targeted therapies that exist or are being developed, although it is briefly mentioned within the text. It may be in the form of table as well.

Author Response

“This article would be of even better quality with the addition of one paragraph that would refer to targeted therapies that exist or are being developed, although it is briefly mentioned within the text. It may be in the form of table as well.”

We have included a new table reporting cytokine targeted therapies to supplement the brief commentary we already provided in the paper. This wonderful change has made the paper more comprehensive. This addition has greatly enhanced the paper.

Reviewer 3 Report

The work is nice and well written. It concerns cytokines, very important element of all processes, (often neglected in articles) in the context of cancers and the connected phenomena. I frankly think it might me published in the present form, however, I propose to introduce two minor changes:

-paragraph 4 is very long, it would be useful to divide it into 2-3 smaller sections

-conclusions - after reading this long article, this sections seems to be not enough - it would be better to expand it a bit

Author Response

“-paragraph 4 is very long, it would be useful to divide it into 2-3 smaller sections”

In response to this suggestion, we have added two sub-sections to the main section 4. This section is now in smaller parts and easier to read because of this great change. The new sub-sections are labelled “4.1 Angiogenesis & The VEGF Family” and “4.2 Angiogenesis & Other Major Cytokines”. We are thankful for this suggestion and how it has improved the paper.

“-conclusions - after reading this long article, this section seems to be not enough - it would be better to expand it a bit.”

We have made additions to the conclusion to increase its length. We agree that lengthening the conclusion is a great idea. Thank you for the suggestion. The additions as shown below in red:

Due to cancer’s global impact, understanding how cancer develops and progresses is critical. Cytokines are heavily involved in regulating cancer developmental processes. Immune activation and suppression, inflammation, cellular damage, angio-genesis, CSCs, invasion, and metastasis are all related to cancer and controlled by cytokines. Cytokines have profound effects on cells and may lead to pro- or anti-tumor activity, depending on environmental conditions and the presence of other cytokines. Overall, studies have shown that cytokines are small messengers that often overlooked as powerful molecular players in the TME. The impact and therapeutic potential of cytokines is beginning to be uncovered through already existing and developing cytokine-targeted therapies. In this review, we have highlighted only a few cytokines and cancer-related processes. Further research focusing on the intricate relationship between cancer and cytokines will be required to increase our understanding of how cancer behaves. Cytokine research will continue to play a key role in revealing the molecular mechanisms behind cancer development and identifying novel therapeutic targets.